# Assessing preferences for HIV pre-exposure prophylaxis (PrEP) delivery services via online pharmacies in Kenya: protocol for a discrete choice experiment

Yilin Chen [1] Enrique M Saldarriaga,[1] Michalina A Montano,[2,3] Kenneth Ngure [4] Nicholas Thuo,[5] Catherine Kiptinness,[5] Maeve Rafferty,[6] Fern Terris-Prestholt [7] Andy Stergachis [1,3] Melissa Latigo Mugambi [3] Katrina F Ortblad [8] Monisha Sharma[3]

For numbered affiliations see end of article.

**Correspondence to**
Yilin Chen; ylchen95@uw.edu

## ABSTRACT

**Introduction** Pre-exposure prophylaxis (PrEP) is highly effective at preventing HIV acquisition, but coverage remains low in high prevalence settings. Initiating and continuing PrEP via online pharmacies is a promising strategy to expand PrEP uptake but little is known about user preferences for this strategy. We describe methods for a discrete choice experiment (DCE) to assess preferences for PrEP delivery from an online pharmacy.

**Methods and analysis** This cross-sectional study is conducted in Nairobi, Kenya, in partnership with MYDAWA, a private online pharmacy retailer with a planned sample size of >400 participants. Eligibility criteria are: ≥18 years, not known HIV-positive and interested in PrEP. Initial DCE attributes and levels were developed via literature review and stakeholder meetings. We conducted cognitive interviews to assess participant understanding of the DCE survey and refined the design. The final DCE used a D-efficient design and contained four attributes: PrEP eligibility assessment, HIV test type, clinical consultation type and user support options. Participants are presented with eight scenarios consisting of two hypothetical PrEP delivery services. The survey was piloted among 20 participants before being advertised on the MYDAWA website on pages displaying products indicating HIV risk (eg, HIV self-test kits). Interested participants call a study number and those screened eligible meet a research assistant in a convenient location to complete the survey. The DCE will be analysed using a conditional logit model to assess average preferences and mixed logit and latent class models to evaluate preference heterogeneity among subgroups.

**Ethics and dissemination** This study was approved by the University of Washington Human Research Ethics Committee (STUDY00014011), the Kenya Medical Research Institute, Nairobi County (EOP/NMS/HS/128) and the Scientific and Ethics Review Unit in Kenya (KEMRI/RES/7/3/1). Participation in the DCE is voluntary and subject to completion of an electronic informed consent. Findings will be shared at international conferences and peer-reviewed publications, and via engagement meetings with stakeholders.

## STRENGTHS AND LIMITATIONS OF THIS STUDY

⇒ This study uses a discrete choice experiment (DCE) to elicit the user preferences on initiating and continuing pre-exposure prophylaxis (PrEP) from an online pharmacy in Kenya.

⇒ DCE results will provide a quantitative assessment of how users value different characteristics of an online PrEP delivery model; this can inform the design of online HIV PrEP delivery programmes to optimise uptake.

⇒ We are collecting information on the willingness-to-pay for each component of online PrEP services, which can inform policy decisions and resource allocation.

⇒ DCE scenarios are hypothetical, and individuals' stated preferences may not represent their actual choices in real-world settings.

⇒ Online PrEP delivery requires access to the internet and ability to pay for HIV self-testing and PrEP delivery services; therefore, this platform is designed for the growing middle class in Kenya, but other PrEP service delivery strategies are needed to reach those with low-income and/or lack of internet access.

## INTRODUCTION

Kenya has one of the largest HIV epidemics globally, with a national prevalence of 4.2%.[1–3] Despite significant strides in HIV prevention, approximately 32 000 new HIV infections occurred in Kenya in 2021.[4] Pre-exposure prophylaxis (PrEP) is highly effective at preventing HIV acquisition and is recommended by the WHO and the Kenya Ministry of Health for persons at risk of HIV.[5] However, PrEP coverage remains low among those with risk indications.[6 7] Currently, PrEP is mainly delivered through healthcare facilities and barriers to widespread use include long wait times and travel distance, limited hours of

operation, privacy concerns, stigma and understaffing.[8 9] Additionally, PrEP demonstration projects in East Africa have found low PrEP retention among those who initiate PrEP at clinics, due in part to challenges returning to the clinic for frequent continuation visits.[10 11]

Initiating and continuing PrEP online is a novel approach that could expand PrEP coverage to individuals not accessing clinic-based PrEP and improve PrEP continuation.[12] Delivery of PrEP services closer to the client can reduce opportunity costs associated with clinic-based PrEP provision. A growing telehealth ecosystem in sub-Saharan Africa, recently expanded due to the COVID-19 pandemic, provides private and convenient options for remote consultation, purchasing and delivery of medications to patients.[13–15] Coupled with Kenya's national commitment to PrEP roll-out, online PrEP delivery is a promising strategy to expand PrEP access. Our team is currently conducting a pilot study of the first online care model for PrEP delivery (NCT05377138) through collaboration with Kenya's first e-pharmacy retailer MYDAWA.[16]

Successful PrEP scale up requires an understanding of user preferences to tailor services to optimise uptake, adherence and retention. There are several potential options for structuring PrEP delivery services via an online pharmacy including the use of different HIV testing modalities and user support options. As online PrEP delivery is not yet available in Kenya, user preferences for components of this service are unknown. Discrete Choice Experiments (DCEs) are an ideal method for assessing user preferences in cases when there is no observed behaviour. With a foundation in economic theory, DCEs assume that individuals choose the service that maximises their expected benefit.[17] DCEs evaluate how individuals value selected features of a service by repeatedly asking them to choose between different sets of hypothetical alternatives. Resulting data can provide a quantitative assessment of user preferences including the relative importance of each attribute and likelihood of service uptake, which can help inform policy decisions and resource allocation. DCEs have been applied to a range of health policy decisions and are increasingly used to assess preferences for PrEP services. Prior DCEs assessing PrEP have examined user preferences for dosing regimen, type of PrEP products, side effects, cost, dispensing venue, and support services.[18–23] However, this is the first study to assess user preferences for online PrEP (ePrEP) delivery.

Here we describe the design and methods from our ongoing DCE to assess user preferences for PrEP delivery from an online pharmacy in Kenya. This study adds to the small but growing literature of DCE protocols for HIV prevention[24–28] and additionally provides detailed information on DCE attribute selection and refinement, materials used for staff training and methods for participant recruitment. The methods described along with slide decks of training materials and survey guides can assist other researchers interested in designing DCEs for HIV prevention. The results of this DCE can inform the design of online PrEP delivery models to increase PrEP uptake among persons at risk of HIV.

## METHODS AND ANALYSIS
### Setting and participants
This study is conducted in Nairobi, Kenya, in partnership with MYDAWA, Kenya's first licensed online pharmacy.[29 30] MYDAWA's online platform provides affordable access to wellness products, prescription and over-the-counter medicines delivered directly to the consumer, including products related to sexual health (eg, HIV self-test kits and emergency contraception). The majority of MYDAWA's customer base is located in Nairobi.[16 29] Our target population is selected to closely mirror persons who would likely be clients of ePrEP delivery. Inclusion criteria are: age ≥18 years, not known to be HIV-positive, interested in PrEP, and able and willing to provide informed consent.

### Patient and public involvement
We developed the care pathway for online PrEP service delivery in collaboration with Kenyan stakeholders, including members from the Ministry of Health, implementing organisations, researchers and professional organisations. We refined the list of attributes and levels via stakeholder discussions with individuals in Kenya with expertise in HIV prevention and PrEP delivery and representatives from MYDAWA. We plan to share study findings with both local and international audiences. We will disseminate results at engagement meetings with key informants including Kenya's National AIDS and STIs Control Programme (NASCOP), patients groups, service providers, and other stakeholders.

### Care pathway for online PrEP service delivery
We collaboratively developed a care pathway for online PrEP service delivery, based on an ongoing model of pharmacy-based PrEP service delivery in Kenya.[31] The core components are: (1) PrEP eligibility assessment: Based on NASCOP PrEP Rapid Assessment Screening Tool, routinely used at public clinics in Kenya to assess HIV risk and PrEP eligibility; (2) HIV testing: Individuals who are potentially eligible for PrEP complete an HIV test to ensure they are HIV-negative prior to starting PrEP; (3) Clinical consultation and PrEP prescribing: Individuals testing HIV-negative undergo a medical risk assessment with a clinical officer to ensure PrEP is safe for them and receive a PrEP prescription; (4) PrEP delivery: A 1-month supply of PrEP is delivered to a setting of the client's choice at initiation. For all future refills, clients receive a 3-month PrEP supply, consistent with the guidelines for PrEP dispensing at public clinics in Kenya. Clients complete the HIV testing process described above prior to each refill visit; and (5) PrEP support: Individuals can opt-in to receive ongoing support to answer questions about PrEP use and to report concerns and side effects. Each step in the care pathway for online PrEP delivery can

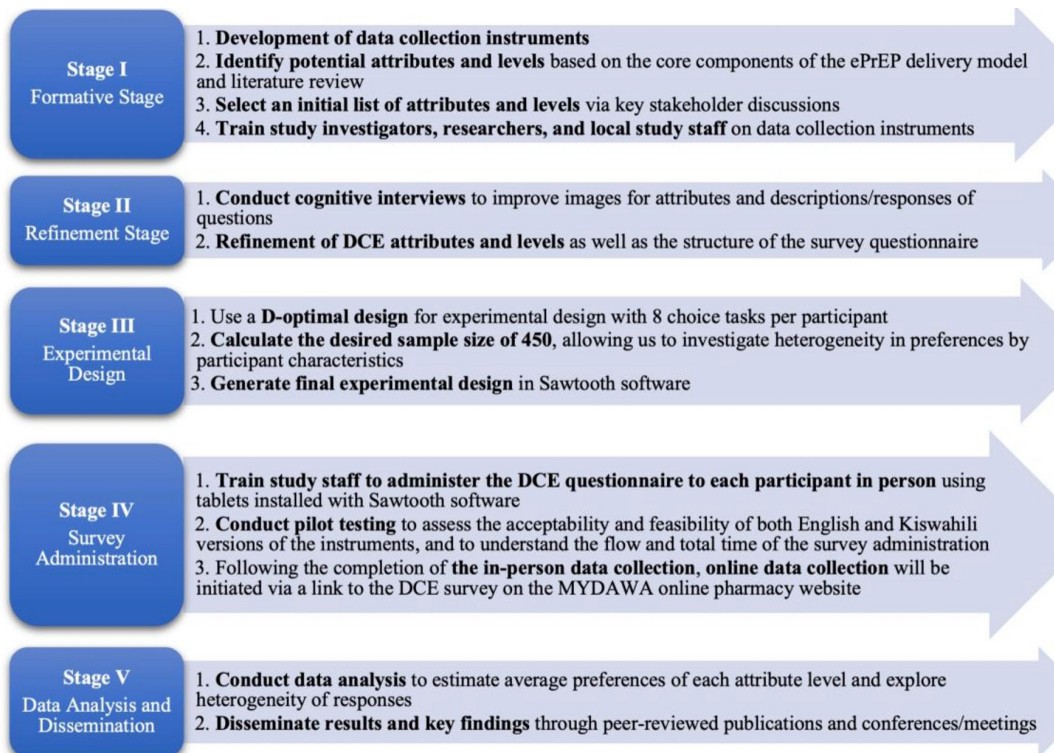

**Figure 1** Study design flowchart. DCE, discrete choice experiment; ePrEP, online PrEP; PrEP, pre-exposure prophylaxis.

be optimised to align with user preferences and was evaluated for inclusion in the final DCE.

### Overview of DCE approach

DCE participants are presented with scenarios consisting of two or more hypothetical services with different characteristics or 'attributes' with varying 'levels' and are asked to choose their preferred services. Surveys generally contain 7–10 scenarios with different levels for each attribute. The experimental design determines the scenario pairs and the series of questions presented to participants. Analysis of DCEs can quantify the impact of each aspect of a service on participants' preference and assess the trade-offs they are willing to make to obtain their preferred service.

Design and administration of the DCE is divided into several stages (figure 1): (1) formative stage, (2) refinement stage, (3) experimental design, (4) survey administration and (5) data analysis and dissemination. First, in the formative stage, we selected attributes and levels for the DCE and developed survey questions to accompany the DCE. Second, in the refinement stage, we conducted cognitive interviews to assess participant understanding of the hypothetical scenarios and associated illustrations (see online supplemental file 1 (cognitive interview guide) and online supplemental file 2 (staff training materials)). We refined our DCE attributes and levels based on participant feedback. In the third stage, we developed the experimental design using a D-optimal design with eight choice tasks per participant. Our study design followed

The Professional Society for Health Economics and Outcomes Research (ISPOR) Good Research Practice for Conjoint Analysis.[32] We piloted the DCE to estimate the time needed to complete the questionnaire and assess participant fatigue. The survey administration adopts a hybrid approach, including in-person administration by research assistants or self-administration by online participants to achieve sufficient sample size and reach our target population more efficiently. Finally, we conduct data analysis to estimate average preferences of each attribute level and explore heterogeneity of responses.

### Software

We use Research Electronic Data Capture (REDCap) electronic data capture tools hosted at the University of Washington for screening and informed consent data.[33] The DCE questionnaire is programmed in Sawtooth Software's Lighthouse Studio modules for general questionnaires and choice-based conjoint scenario designs, and hosted on the Sawtooth servers.[34] We contacted a Sawtooth representative and obtained the software at a discounted price. Screening and informed consent forms are constructed as separate, chained surveys in REDCap. Completion and submission of informed consent forms via REDCap directs online participants to the Sawtooth questionnaire via an external link for online participants. DCE questionnaires are administered via Sawtooth's offline data capture application for in-person participants.

**Table 1** DCE attributes and levels

| Attribute and description | Levels |
|---|---|
| PrEP eligibility assessment (Method for conducting client eligibility assessment for PrEP) | 1. Online self-assessment using screening questions (phone number provided in case of questions) |
| | 2. Guided assessment with a remote clinical provider (via a phone call or WhatsApp) |
| HIV test type (Type of HIV test delivered for PrEP initiation) | 1. Oral fluid HIV self-test (at setting of your choice) |
| | 2. Blood-based HIV self-test (at setting of your choice) |
| | 3. Provider administers HIV Test at setting of your choice (blood-based) |
| Clinical consultation for prescribing PrEP (Consultation needed to prescribe PrEP) | 1. Remote clinical consultation with provider (via a phone call or video chat) |
| | 2. In-person clinical consultation with provider after completing HIV test (at a setting of your choice) |
| User support options for PrEP (Method for discussing your questions for PrEP with a healthcare provider) | 1. Short messaging service |
| | 2. Phone/video call |
| | 3. Email |

DCE, discrete choice experiment; PrEP, pre-exposure prophylaxis.

## DCE procedures
### Stage I: formative phase—selection of attributes and levels, development of data collection instruments, training
*Collaborative attribute and level selection*

We developed an initial list of attributes and levels based on the care pathways of the ePrEP delivery model and a review of the PrEP literature.[5 7 35–37] This list was refined during stakeholder discussions with individuals in Kenya with expertise in HIV prevention and PrEP delivery and representatives from MYDAWA. Five attributes were included in the formative list: PrEP eligibility assessment, HIV test type, clinical consultation, user support options and cost of PrEP delivery. Each attribute consists of two to three levels (table 1).

*Questionnaire development*

We developed a survey to accompany the DCE which includes the following: PrEP eligibility assessment, PrEP interest and knowledge, online pharmacy and HIV testing interest, knowledge and engagement, and participant demographics, and sexual behaviour (see online supplemental file 3 for survey questionnaire). We also assessed maximum willingness to pay (WTP) for different components of online PrEP delivery including both oral and blood-based HIV self-testing, remote clinical encounter and delivery of PrEP drugs. The

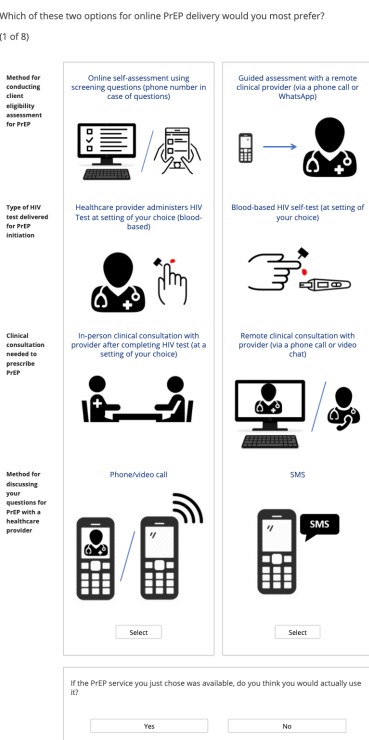

**Figure 2** Example of DCE choice task. DCE, discrete choice experiment; PrEP, pre-exposure prophylaxis; SMS, short messaging service.

structure of the WTP questions was informed by a literature review.[38–40] The DCE begins with a brief overview of DCEs, definition of attributes and levels. Participants then complete the choice tasks. For each DCE scenario, participants are first asked to choose their preferred service followed by an opt-out question: 'Would you choose to get PrEP using this service if it were available?' (figure 2).

*Staff training*

We conducted a training with study investigators, data mangers and in-country research staff responsible for administering the survey; it was held over Zoom split over two half-day sessions to accommodate the time difference between Seattle and Kenya. The training consisted of presentations on the background of HIV in Kenya, the role of PrEP for HIV prevention, barriers to accessing PrEP via current delivery modalities and the potential ePrEP delivery to increase PrEP coverage. On the first day, we explained the DCE administration, including the REDCap pre-screening form, process for scheduling in-person visits, REDCap e-consent forms, and the Sawtooth survey. On the second day, we discussed the process for conducting cognitive interviews and piloting and launching the DCE. Study staff practiced administering the cognitive interviews and DCE instruments in groups of two and provided feedback on study instruments (see online supplemental file 2 for slide deck and training materials).

## Stage II: refinement of DCE attributes and levels, and cognitive interviews
### DCE image design and cognitive interviews

We used images to pair with each level of DCE attributes to make the survey more engaging and understandable. Most of the images were adapted from the Noun Project, a free online image bank.[41] We chose simple images to minimise the possibility of biasing participants. For example, images of healthcare providers are depicted with neutral expressions and their gender is not specified. In addition, images of fingerpicks for HIV testing scenarios are not portrayed with big needles or large amounts of blood.

### Cognitive interviews

We recruited individuals from the HIV research community in Kenya for cognitive interviews to assess the clarity of DCE images, scenarios and WTP questions (see online supplemental file 1 for cognitive interview guide). Participants were asked to interpret each attribute image and assess its accuracy in conveying the specified DCE level. They were then presented with alternative images for each attribute and asked if any were a better match for the attribute description than the current image. The cognitive interviews also presented participants with two example DCE choice tasks to assess their understanding of the scenarios and ease of completion. Additionally, we assessed participants' understanding of WTP questions. We initially sought to audio-record the cognitive interviews but after consultation with the in-country staff, decided that it may hinder participants' ability to speak freely; therefore, study staff took notes during the interviews.

### Stage III: experimental design and pilot testing

We used a D-efficient design to construct the DCE.[32] The algorithm behind this design identifies the subset of the most meaningful comparisons across profiles from all possible choice tasks—1296 (ie, $(2 \times 3 \times 2 \times 3)^2$ in our study.[42] The design enhances data efficiency and attempts to balance statistical power and sample size.[43 44] First, we considered the complete specification of the utility functions based on the chosen attributes and levels and multinomial logit as the analytical model.[42] We defined a main-effects utility function for each of the two alternatives in a choice task. The structure for the $i$-th utility function, $U$ is defined as:

$$U_i = V(\beta, X_i) + \epsilon_i$$
$$V(\beta, X_i) = \beta_1 * X_1 + \cdots + \beta_k * X_k$$

Where $V(\beta, X_i)$ is the deterministic portion of the utility and a function of the $k$ attribute-levels considered in the alternative; the $\beta$ coefficients represent the average preference weight that the sample places on each attribute-level. The error term, $\epsilon_i$, indicates the inability to perfectly measure the utility of each alternative, which is consistent with random utility theory. We did not use priors for the utility-function coefficients since none of the selected attributes had a natural order (ie, an ascending order example is 'pain' as we assume that most people prefer low to high pain).

Each scenario consists of a forced-choice task, followed by an opt-out question: 'Would you choose to get PrEP using this service if it were available?' (figure 2).[45] The goal of the opt-out option was to represent the status quo where participants can decide against uptake of a new product.[46] This format allows us to gather data about user preferences while not drawing too much attention to the opt-out. Studies suggest potential for cognitive biases to preserve the status quo in three-alternative choice tasks.[47 48]

### Sample size

The DCE literature has not yet reached a consensus on the best method to estimate the sample size required to return meaningful, statistically robust parameter estimates.[49 50] Some experts suggest using a minimum sample size of 300.[49] Johnson and Orme provide another rule of thumb using an equation that considers the number of choice questions, alternative scenarios presented and attributes and levels.[50 51] Additionally, the more tasks per participant, the smaller sample size required to achieve comparable levels of uncertainty in coefficients. During the formative phase of our study, most participants stated that they could answer between 7 and 10 choice tasks before becoming fatigued.

A criticism to rules of thumb for DCE sample estimation is the lack of robustness to different product characteristics or consumers behaviour.[52] Given this uncertainty, we implemented a scenario analysis considering all possible tasks per participant, from 6 to 10 with a sample size varying from 200 to 500 in 50 increments. For each scenario, we evaluated the standard error (SE) of coefficients and the D-efficiency measure for the overall design. We found that improvements were diminishing above a sample size of 350. Therefore, we chose a design with 8 choice tasks per participant and a sample size of 400, which had an expected standard deviation (SD) of 0.049 per coefficient. This sample size was slightly higher than 350 to account for the possibility of preference heterogeneity and incomplete DCE surveys.[32 53]

### Stage IV: pilot testing, DCE recruitment and administration
### Pilot testing and survey refinement

After revising the DCE survey based on feedback from the cognitive interviews, we piloted it in two phases. First, we administered the survey to individuals from the HIV research community in Kenya. Based on participants' feedback, we found the cost attribute seemed to be a strong driver of participant preferences, with many participants selecting the cheaper PrEP service regardless of other attributes. To avoid the possibility of having one attribute overwhelming participant decision-making, we removed the cost attribute from the final version of the DCE and instead assessed cost preferences only through direct WTP questions (table 1). Additionally, we moved the DCE scenario questions to the beginning of

the survey to minimise participant fatigue. Since several participants expressed hesitancy in answering sensitive sexual behaviour questions at the beginning of the survey, we moved these questions to the end to provide increased opportunity for staff to develop rapport with participants; we also gave participants the option of self-administering these questions by taking the tablet and filling out this section on their own. Finally, participants found the DCE eligibility questions from the HIV Risk Assessment Tool to be invasive and too sensitive to answer over the phone, so we revised the pre-screening process to only ask if participants are interested in obtaining PrEP. In the second phase, we piloted the survey to 20 individuals recruited through the MYDAWA platform using study eligibility criteria. We assessed data quality, survey flow and time needed for survey administration.

### Recruitment

Similar to other DCEs in the literature, we use online recruitment strategies.[23][24] Participants are recruited through banner advertisements on the MYDAWA online pharmacy website. To target persons at risk of HIV acquisition, only those who click on HIV self-test kits on the MYDAWA platform see the online recruitment banners. In addition, persons who purchase HIV self-test kits (HIVSTs) from MYDAWA are mailed a flyer with their purchases that advertises the DCE (see online supplemental file 4 for the recruitment banner). We are also planning to expand DCE recruitment by advertising on social media pages that provide information on HIV prevention as well as through the MYDAWA call in centre where persons can access support related to sexually transmitted infection purchases.

### Survey administration

Interested participants can voice call or send an short messaging service to the study phone number provided on recruitment materials. Study staff assess eligibility via a phone pre-screening and schedule time to meet participants at a convenient location to complete the survey (see online supplemental file 5 for the pre-screening script). Participants are reimbursed 1000 Kenyan shilling (US$8.70) for their time. The in-person data collection was initiated on 1 June 2022 and expected to be completed by 28 February 2023. We plan to launch the online self-administered DCE after we finish data collection for the in-person component. For the online DCE, participants will follow the DCE link on the recruitment banner to an online screening tool, self-administered via REDCap survey. Once the screening survey is submitted, eligible participants will automatically be directed to complete online consent forms via REDCap, and subsequently the DCE questionnaire via Sawtooth.

### Stage V: data management and data analysis

Data are stored in a central location on the Sawtooth administration site which allows users to examine survey responses or download a csv file for further analysis.

We monitor data quality on a weekly basis, including checking number of interviews, duplicate entries, incomplete surveys, and substantial missingness or outliers in responses (see online supplemental file 6 for R script used for data management).

To analyse the data, we will fit a conditional logit model with an alternative specific constant for the opt-out. The coefficients (ie, preference weights) represent the average relative utility for each attribute-level, compared with the reference.[54][55] In addition, we will generate a ranking of importance across attributes calculated as the ratio of their preference weights over the preference weights of the attribute with the lowest impact in the decision-making process (ie, the attribute with the lowest preference weights). This ranking will provide information about the relative importance of each attribute, and respondents' willingness to trade one attribute for another. We will conduct a two-part analysis for the main model: First, we will analyse only the forced-choice data (ie, ignoring the opt-out question), and second, we will use a mixed logit regression model (ie, random parameters logit) to account for preferences' heterogeneity in all coefficients. We will qualitatively assess if our conclusions are consistent in both analyses and compare the goodness of fit of the models using Akaike information criteria (AIC) and Bayesian information criteria (BIC).

Further, we will elicit preference heterogeneity by participant demographics and other characteristics. Latent class analysis will be performed to examine variation in preference by using the data to identify groups of respondents with similar preferences.[56] We will fit 2–10 classes and assess the goodness of fit of each one using McFadden's pseudo $R^2$,[54] log likelihood test and adjusted-AIC and adjusted-BIC. In a scenario in which a covariate is an important predictor of class membership, we will assess its potential impact as an effect modifier and assess the results separately for each category of the covariate.

Descriptive analysis will be conducted to summarise participants' demographic characteristics, sexual behaviours, WTP for online PrEP services, and online pharmacy interest/engagement. Data analysis will be conducted using Sawtooth's built-in analysis capabilities and R Version 4.1.2.

### ETHICS AND DISSEMINATION
### Ethics considerations

This study was approved by the University of the Washington Human Research Ethics Committee (STUDY00014011), the Kenya Medical Research Institute (KEMRI), Nairobi County (EOP/NMS/HS/128) and the Scientific and Ethics Review Unit (SERU) in Kenya (KEMRI/RES/7/3/1). Participation in the DCE is voluntary and subject to completion of an electronic informed consent. Respondents' personal information is kept secure and confidential. Participants are not required to use their real names and we do not collect any identifiable

information. Participants are told that they can withdraw from the survey at any time.

## Dissemination

Findings will be shared at international conferences and peer-reviewed publications. We will disseminate results at engagement meetings with key informants including Kenya's NASCOP and other stakeholders. Preliminary results will be used to inform the design of the ePrEP Kenya Pilot study (NCT05377138) which will evaluate uptake and continuation rates for online PrEP delivery via MYDAWA in Nairobi, Kenya. Our results can be used to design online PrEP delivery models to optimise PrEP uptake in Kenya and similar settings.

#### Author affiliations
[1]The Comparative Health Outcomes, Policy, and Economics (CHOICE) Institute, University of Washington, Seattle, Washington, USA
[2]Vaccine and Infectious Diseases Division (VIDD), Fred Hutchinson Cancer Center, Seattle, Washington, USA
[3]Department of Global Health, University of Washington, Seattle, Washington, USA
[4]School of Public Health, Jomo Kenyatta University of Agriculture and Technology, Nairobi, Kenya
[5]Partners in Health Research and Development, Center for Clinical Research, Kenya Medical Research Institute, Nairobi, Kenya
[6]MYDAWA, Nairobi, Kenya
[7]Department of Global Health and Development, London School of Hygiene and Tropical Medicine Faculty of Public Health and Policy, London, UK
[8]Public Health Science Division, Fred Hutchinson Cancer Center, Seattle, Washington, USA

**Acknowledgements** We would like to thank Peter Mogere from KEMRI and Daniel Were from Jhpiego for participating in stakeholder discussions. We thank Tony Wood, Lisbet Charana and other MYDAWA staff for their support of this project and assistance with promoting the study on their website. We would like to acknowledge Dr Nicole Young (Program Officer, New Product Planning & Introduction, HIV/TB) and her colleagues at the Bill & Melinda Gates Foundation for their support in developing and funding this work (Grant #: INV-037646). We are grateful to study staff and survey participants.

**Contributors** MS, KFO and MLM conceived the study. YC, EMS, MAM, KN, NT, CK, MR, FT-P, AS, MLM, KFO and MS contributed to its design. YC, EMS, MAM and MS wrote the first draft of the protocol and designed the discrete choice experiment tasks. All authors contributed to the drafting and editing of the manuscript, and approved the final version.

**Funding** This study was funded by the Bill & Melinda Gates Foundation (Grant #: INV-037646). The funders had no role in study design, data collection, analysis, writing of the report nor the decision to submit for publication.

**Competing interests** None declared.

**Patient and public involvement** Patients and/or the public were involved in the design, or conduct, or reporting, or dissemination plans of this research. Refer to the Methods section for further details.

**Patient consent for publication** Not applicable.

**Provenance and peer review** Not commissioned; externally peer reviewed.

#### ORCID iDs
Yilin Chen http://orcid.org/0000-0003-0040-3881
Kenneth Ngure http://orcid.org/0000-0002-8062-0933
Fern Terris-Prestholt http://orcid.org/0000-0003-1693-5196
Andy Stergachis http://orcid.org/0000-0003-0057-6627
Melissa Latigo Mugambi http://orcid.org/0000-0001-8875-5528
Katrina F Ortblad http://orcid.org/0000-0002-5675-8836

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
