## [Reviewer comments · BMJ Open]

ARTICLE DETAILS

TITLE (PROVISIONAL)	Assessing preferences for HIV pre-exposure prophylaxis (PrEP) delivery services via online pharmacies in Kenya: protocol for a discrete choice experiment
AUTHORS	Chen, Yilin; Saldarriaga, Enrique; Montano, Michalina; Ngure, Kenneth; Thuo, Nicholas; Kiptinness, Catherine; Rafferty, Maeve; Terris-Prestholt, Fern; Stergachis, Andy; Mugambi, Melissa; Ortblad, Katrina; Sharma, Monisha

VERSION 1 – REVIEW

REVIEWER	Wasim Baqir Northumbria Healthcare NHS Foundation Trust, Pharmacy
REVIEW RETURNED	21-Dec-2022

GENERAL COMMENTS	I have used this paper by Mangham et al to help me review this manuscript. https://academic.oup.com/heapol/article/24/2/151/591265?login=false The protocol is well written and the key aspects of designing a Discrete Choice Experiment have been covered. Some points for the authors to consider: - Attributes (user support): have you considered digital-poor people or are they excluded from the study?- Recruitment (page 19) - how does this compare from other DCE research?
---

REVIEWER	Cassidy Claassen University of Maryland School of Medicine, Center for International Health, Education, and Biosecurity
REVIEW RETURNED	30-Jan-2023

GENERAL COMMENTS	PrEP is an increasingly important aspect of HIV prevention. To achieve further scale up, PrEP access needs to be decentralized and brought into pharmacies at all levels. There is limited information how to best achieve this; this study aims to fill that gap by conducting discrete choice experiments around user preferences, particularly in regards to online pharmacies. The Methods seem sound, however I should note that I am not an expert in DCE. The significance of the findings however will be important for PrEP scale up in sub-Saharan Africa, and I commend the authors for taking on this ambitious project.
--

	Recommendations: 1. Ensure that the protocol is reviewed by someone with DCE expertise. 2. Recommend acceptance following the above.
--	--

VERSION 1 – AUTHOR RESPONSE

Reviewer: 1

Dr. Wasim Baqir, Northumbria Healthcare NHS Foundation Trust

Comments to the Author:

I have used this paper by Mangham et al to help me review this manuscript.

<https://academic.oup.com/heapol/article/24/2/151/591265?login=false>

The protocol is well written and the key aspects of designing a Discrete Choice Experiment have been covered.

Response: We thank the author for their positive feedback.

Some points for the authors to consider:

- Attributes (user support): have you considered digital-poor people or are they excluded from the study?

Response: We thank the reviewer for this question. The online PrEP delivery intervention (which our DCE is assessing preferences for) requires access to internet and ability to pay for HIV self-testing and PrEP delivery services; therefore, this platform is targeted to reaching the growing middle class in Kenya but is not accessible to those with low socioeconomic status. Therefore the reviewer is correct that digitally poor people would not be likely to access the intervention. Different interventions are needed to target those without internet access. We have included this as a limitation of the study in the 'Strengths and limitations of this study' section. We plan to further discuss this important point in our main publication of DCE results. While we do not explicitly exclude participants without internet access from our study, our recruitment strategy targets those who would be most likely to be users of the online PrEP service delivery model. Participants recruited through banner ads on the MYDAWA

online pharmacy website or through flyers mailed to prior clients, which makes sure that the participants we reached are internet savvy. These recruitment strategies were intentionally chosen as DCEs should be designed to enroll the target population who would access the intervention, as their preferences are most relevant to inform optimal program design.

- Recruitment (page 19) - how does this compare from other DCE research?

Response: We developed the recruitment strategy through discussion with in-country stakeholders. Since the target population for our study is selected to closely mirror persons who would likely be clients of ePrEP delivery, we recruited participants through banner ads on the MYDAWA online pharmacy website. In addition, persons who purchase these items from MYDAWA are mailed a flyer describing the DCE with their purchases. This is the first DCE study eliciting preferences for online PrEP delivery so it is not directly comparable to other studies, however many DCEs in the literature conduct online recruitment to enroll participants. We now specify this in the manuscript:

Page 18:

“Similar to other DCEs in the literature, we utilize online recruitment strategies.”

Reviewer: 2

Dr. Cassidy Claassen, University of Maryland School of Medicine

Comments to the Author:

PrEP is an increasingly important aspect of HIV prevention. To achieve further scale up, PrEP access needs to be decentralized and brought into pharmacies at all levels.

There is limited information how to best achieve this; this study aims to fill that gap by conducting discrete choice experiments around user preferences, particularly in regards to online pharmacies.

The Methods seem sound, however I should note that I am not an expert in DCE.

The significance of the findings however will be important for PrEP scale up in sub-Saharan Africa, and I commend the authors for taking on this ambitious project.

Recommendations:

1. Ensure that the protocol is reviewed by someone with DCE expertise.
2. Recommend acceptance following the above.

Response: We thank the reviewer for their positive comments.